# PeerJ

# How male sound pressure level influences phonotaxis in virgin female Jamaican field crickets (*Gryllus assimilis*)

Karen Pacheco and Susan M. Bertram

Department of Biology, Carleton University, Ottawa, ON, Canada

## ABSTRACT

Understanding female mate preference is important for determining the strength and direction of sexual trait evolution. The sound pressure level (SPL) acoustic signalers use is often an important predictor of mating success because higher sound pressure levels are detectable at greater distances. If females are more attracted to signals produced at higher sound pressure levels, then the potential fitness impacts of signalling at higher sound pressure levels should be elevated beyond what would be expected from detection distance alone. Here we manipulated the sound pressure level of cricket mate attraction signals to determine how female phonotaxis was influenced. We examined female phonotaxis using two common experimental methods: spherical treadmills and open arenas. Both methods showed similar results, with females exhibiting greatest phonotaxis towards loud sound pressure levels relative to the standard signal (69 vs. 60 dB SPL) but showing reduced phonotaxis towards very loud sound pressure level signals relative to the standard (77 vs. 60 dB SPL). Reduced female phonotaxis towards supernormal stimuli may signify an acoustic startle response, an absence of other required sensory cues, or perceived increases in predation risk.

## INTRODUCTION

Acoustic communication in many anurans and insects is an essential part of courtship and reproductive behaviour (*Alexander, 1975*; *Wells, 1977*; *Gerhardt, 1991*; *Gerhardt & Huber, 2002*). Acoustic sexual signalling serves to attract receptive females from a distance (*Walker, 1957*; *Alexander, 1962*; *Gerhardt, 1991*; *Ryan & Keddy-Hector, 1992*). Behavioural studies have identified several specific parameters of male acoustic signals used for species recognition and to confer attractiveness (*Rand & Ryan, 1981*; *Gerhardt, 1982*; *Simmons & Zuk, 1992*; *Wilczynski, Rand & Ryan, 1995*), with a broad assessment of the literature revealing that females generally prefer more energetic signallers (reviewed by *Ryan & Keddy-Hector, 1992*).

Sound pressure level (SPL) can be an important predictor of mate choice (*Stout & McGhee, 1988*; *Castellano et al., 2000*; *Klappert & Reinhold, 2003*; *Hedwig & Poulet, 2005*) with females tending to prefer more intense signals when given the choice

Corresponding author
Susan M. Bertram,
sue.bertram@carleton.ca

**Table 1 Variation in the sound pressure levels used in female field cricket preference studies.**

| Study | Species | SPL (dB) | Speaker distance (cm) |
|---|---|---|---|
| *Scheuber, Jacot & Brinkhof, 2004* | *Gryllus campestris* | 80[*] | 10.0 |
| *Lickman, Murray & Cade, 1998* | *Gryllus texensis* | 82–85[*] | 35.4 |
| *Bailey, 2008* | *Teleogryllus oceanicus* | 70 | |
| *Prosser, Murray & Cade, 1997* | *Gryllus texensis* | 83[*] | 35.4 |
| *Wagner, Murray & Cade, 1995* | *Gryllus texensis* | 84.6 | 35.5 |
| *Hedrick, Hisada & Mulloney, 2007* | *Gryllus integer* | 72 | |
| *Olvido & Wagner, 2004* | *Allonemobius socius* | $66 \pm 2$ | 5.0 |
| *Gray & Cade, 2000* | *Gryllus texensis* | 84 | |
| *Gray & Cade, 2000* | *Gryllus rubens* | 84 | |

**Notes.**

[*] Reference is made to the natural range of sound pressure levels.

(*Whitney & Krebs, 1975*; *Sullivan, 1983*; *Wells & Schwartz, 1984*; *Ryan, 1985*; *Gerhardt, 1991*). For example, *Arak (1988)* revealed that female natterjack toads (*Bufo calamita*) can perceive small differences in male sound pressure levels and typically prefer males that signal at higher sound levels. Female wax moths (*Achroia grisella)* also prefer males that produce ultrasound signals that contain higher acoustic energy and power (*Jang & Greenfield, 1996*).

Males that produce signals at a higher sound pressure level should be detectable at greater distances; males that signal with high sound pressure level should therefore receive a selective advantage over males that signal with low sound pressure level and subsequently have a shorter detection distance (*Forrest & Green, 1991*; *Forrest & Raspet, 1994*). However, if females also prefer higher sound pressure level males over lower sound pressure level males then the potential fitness impacts of higher sound pressure level signalling should be elevated beyond what would be expected from detection distance alone. It is therefore important to quantify female preference functions for sound pressure level to gain insights into the evolutionary consequences and patterns of selection on this trait.

Female preference functions for sound pressure level have been understudied in crickets despite extensive studies on female preference for other male signal components. Adult male field crickets produce long distance acoustic mate attraction signals (also known as calls) by rubbing their modified forewings together (*Alexander, 1962*). When a male closes his wings, the scraper of one wing hits the teeth of the file on the other wing causing the harp to resonate and produce a single pulse of sound (*Bennet-Clark, 2003*). Males concatenate these pulses into chirps (*Alexander, 1962*; *Bennet-Clark, 2003*) and females use these long distance acoustic signals to both orient towards signalling males (phonotaxis) and choose between potential mates (*Alexander, 1962*). To date, only a handful of the experiments quantifying female cricket preference for sound pressure level have explored a broad range of sound levels (Table 1). Even fewer have explored the extremes of the natural range of sound pressure levels available to females. Further, most studies examining female preference for sound pressure level often simultaneously manipulate other signalling

parameters, making it difficult to tease apart the influence that sound pressure level alone has on female preference. For example, *Stout & McGhee (1988)* examined the relative importance of pulse + interpulse duration (syllable period), chirp rate and signal sound pressure level on female mate preference. Unfortunately, *Stout & McGhee (1988)* made no reference to the natural range of sound pressure levels found in nature. Instead, they quantified female phonotactic response at a 2 dB, 5 dB and 10 dB SPL increase above their standard male signal of 65 dB SPL. By presenting female European house crickets (*Acheta domesticus*) with a pair of male signals differing in one or more of these parameters, *Stout & McGhee (1988)* concluded that sound pressure level was more important than chirp rate, but that syllable period was more important than sound pressure level in influencing female mate choice. While there is emerging interest in determining how multiple signal parameters interact to influence patterns of selection and the evolutionary consequences of signalling (*Brooks et al., 2005*), it is first worth examining how sound pressure level alone shapes female preference functions.

Here we quantify female Jamaican field crickets (*Gryllus assimilis*) phonotactic response to variation in male long distance acoustic mate attraction signal sound pressure level using two standard research methods, a spherical treadmill and open arena. These two methods fall within two broader categories of testing: open-loop and closed-loop methods, respectively. Open-loop methods (Kramer spherical treadmill also called a trackball or locomotion compensator) tether a female in one place, so that she does not experience changes in sound pressure level as she walks 'toward' the mate attraction signal (*Kramer, 1976*; *Weber, Thorson & Huber, 1981*; *Doherty, 1985*; *Doherty & Pires, 1987*; trackball: *Hedwig & Poulet, 2005*; *Hedwig, 2006*). Closed-loop methods (open arena) allow the female to move within the enclosed space, thereby allowing her to experience natural changes in sound pressure level as she approaches the mate attraction signal. Quantifying female preference for the sound pressure level of the acoustic mate attraction signal provides a powerful comparison of whether one technique quantifies female phonotaxis more effectively than the other because sound pressure level is one of the primary components that differ between the two techniques. The focus of our study was therefore two-fold: (1) to examine female phonotaxis toward long distance acoustic mate attraction signals that vary in sound pressure level across the natural range of this species, and (2) to compare female phonotaxis on the spherical treadmill to the open arena.

## METHODS

### Cricket rearing

Our founding population of *Gryllus assimilis* was originally collected in Bastrop County, Texas, United States (latitude ∼ 30°17′N, longitude ∼ 97°46′W, elevation ∼ 145 m) from 15 to 24 September, 2008. We did not require specific permits for collecting invertebrates because these crickets are neither endangered nor protected. We imported adult crickets and eggs to the greenhouse laboratories at Carleton University, Ottawa, Canada (Canadian Food Inspection Agency permit #2007-03130). Our greenhouse facilities are Plant Pest Containment Level 1 certified (Canadian Food Inspection Agency permit #P-2012-03836).

Our study was conducted in accordance with the guidelines of the Canadian Council on Animal Care.

The crickets were reared in communal plastic bins ($L \times W \times H = 64$ cm $\times$ 40 cm $\times$ 42 cm) with a 14:10 h $L : D$ illumination period (lights on at 0600 h, off at 2000 h) at $28 \pm 2$ °C. They were fed *ad libitum* food (Harlan Teklad Rodent diet 8604M, Harlan Laboratories, Indianapolis, IN, USA: 24.3% protein, 40.2% carbohydrate, 4.7% lipid, 16.4% fiber, 7.4% ash). Water was provided in plastic containers filled with marbles to provide a surface to perch on to minimize drowning. In late 2012 (12–16 generations after field collection) we haphazardly collected 30 4th instar females from the communal cricket bins (no wing bud development; ovipositor had just become visible). These juveniles were housed together in a separate communal bin (same conditions as described above) and monitored daily for imaginal eclosion. Within 24 h of imaginal eclosion we transferred the adult females to individual 520 mL ($D \times H = 11$ cm $\times$ 7 cm) clear plastic circular containers with screened lids (4 cm $\times$ 4 cm). Each female was provided with a small piece of cardboard egg container for shelter, *ad libitum* food, and 4–5 water gels every 2–3 days (created by soaking 2–4 mm sized water polymer crystals in distilled water for 4–6 h; Feeder Source, Cleveland, GA, USA). The light cycles and temperatures were identical to the communal rearing environment.

## Standard and focal mate attraction signals

We created our standard and focal mate attraction signals using Adobe Audition CS5.5 software (Adobe Systems Incorporated, San Jose, California, USA). We fashioned our signals after results published in *Whattam & Bertram (2011)*. Because Whattam and Bertram's (*2011*) study revealed that long distance acoustic mate attraction signaling was influenced by diet, we selected our signals' parameters to reflect the mean from a population of males reared on high quality food (recordings made at 26 °C). The signaling parameters we used for both standard and focal signals were: carrier frequency = 3,719 Hz, pulse duration = 10.14 ms, interpulse duration = 15.21 ms, pulses per chirp = 8, and interchirp duration = 1,055 ms (signal available at Figshare DOI http://dx.doi.org/10.6084/m9.figshare.1037378). *Whattam & Bertram (2011)* revealed that well-fed male crickets signal at $60.60 \pm 8.45$ dB SPL ($\tilde{X} \pm 1$ SD), ranging 34–72 dB SPL. We therefore always set our standard signal to broadcast at 60.6 dB SPL (re: 20 μPa RMS). We used four focal sound pressure levels which we labelled very quiet, quiet, loud, and very loud, broadcast at 43 dB $\cong \tilde{X} - 2$ SD, 52 dB $\cong \tilde{X} - 1$ SD, 69 dB $\cong \tilde{X} + 1$ SD, and 77 dB $\cong \tilde{X} + 2$ SD SPL, respectively. Our focal sound pressure levels thus spanned the natural range and slightly beyond that which females are accustomed to hearing. We measured sound pressure level using an EXTECH Digital Sound Level Meter (Model #407768; FLIR Systems, Waltham, MA, USA).

### Preference trials

Each female's phonotaxis was tested using two-choice trials where the standard signal was simultaneously presented against one of the four focal signals. Each female was tested across all four focal signals in both the arena and the spherical treadmill, for a total of

8 tests. These comparisons occurred across four consecutive days, 10–13 days post imaginal moult (hereafter referred to as 10–13 days old) because this is the age range when female *G. assimilis* are most phonotactically responsive (*Pacheco et al., 2013*). On a single test day, an individual female was tested once on the spherical treadmill and once in the arena, each to randomly assigned focal signals. Females were always given at least 1 h of rest prior to switching methods. The order of the method the female was tested on first was randomized.

In both spherical treadmill and open arena trials females were given 60 s to acclimatize in silence (detailed below). Each signal was then broadcast on its own for 30 s (order and speaker side randomized) to ensure the female heard both the focal and standard signals prior to starting the trial. Both signals were then broadcast simultaneously from the speakers with the signal chirps interleaved (alternating), so that a female had the potential to identify which signal was coming from which speaker. Once both focal and standard signals were being broadcast, the trial began and female phonotaxis was recorded (detailed below). Every trial ran a total of 5 min. All trials were run in the evening (between 5:00 PM and 8:00 PM) under a 75 wattage red light.

### Spherical treadmill preference trials

We ran all spherical treadmill trials in a chamber ($L \times W \times H = 86 \times 87 \times 57$ cm) lined with sound-attenuating foam. The spherical treadmill was located in the middle of the chamber with two speakers on either side of it, each 17 cm from the center of the sphere and directly across from each other. Prior to mounting the female onto the spherical treadmill we ensured the standard and focal signal would broadcast at the correct sound pressure level by pointing the sound level meter probe directly at the active speaker, directly above the spherical treadmill, 17 cm from the active speaker, and setting the volume of the standard or focal signal accordingly.

To mount the female on the spherical treadmill we attached a coil- (micro-compression) spring (diameter: 3 mm, length: 8 mm; spring constant: 210.15 N/m) to each female's pronotum using low melting point wax. Coil springs were attached on day 9 post final moult (one day prior to the first trial, *sensu Pacheco et al., 2013*). We mounted each focal female on the treadmill by attaching her spring to a magnetic rod above the treadmill. This mounting ensured that while the female cricket was held firmly in place on the top of the polystyrene ball, she could freely turn 360 degrees, as well as walk or run using natural motions (photos in *Pacheco et al., 2013*). We adjusted the air pressure flowing to the polystyrene ball such that, with the cricket in position, the ball was able to rotate freely in all directions in response to the cricket's walking, running, or turning movements. We oriented all females in the same neutral position at the start of their trial: tethered on the spherical treadmill facing straight ahead between the two speakers.

Once the female was mounted on the spherical treadmill, but prior to the start of data collection, the female experienced 60 s of silence, then 30 s of the focal sound and 30 s of the standard sound (order and speaker side randomized). The female was free to walk during this acclimatization period. We then broadcast both the focal and
standard signals, with the signal chirps interleaved. Data collection on female phonotaxis began as soon as both the focal and the standard signals were being broadcast. Female phonotaxis was recorded relative to the focal speaker. Both standard and focal signals were presented continuously throughout the 5 min trial, and the female's locomotor behavior was recorded from the treadmill at a sample rate of 20 samples per second. Each 5 min trial yielded a total of 6000 samples of cricket $X, Y$ positions. Temperature was held at 22–23 °C in the trial room and was monitored with a Fisher Scientific Traceable Digital Thermometer (Model #15-077-20; Fisher Scientific, Toronto, Ontario, Canada).

We calculated instantaneous displacement (cm) and velocity (cm/s) vectors from the positional data ($X, Y$ coordinates). Total path length was calculated as the sum of all vector lengths over the 6000 samples. Female preference was quantified using net vector score (after *Huber et al., 1984*). Net vector score is the female's net movement toward or away from the focal signal during the 5 min trial and takes into account the female's direction (vector angle) and the vector length of every recorded leg movement:

$$\text{Net Vector Score} = \sum_{t=1}^{6000}[\cos(\text{vector angle}(t)) \times \text{vector length}(t)].$$

We defined the angle of the focal speaker as 0°. Females moving directly toward the focal speaker (0°) had positive vector scores [$\cos(0°) = 1$], females moving directly away from the focal speaker (180°) had negative vector scores [$\cos(180°) = -1$], and females moving perpendicular to the focal speaker (90° or 270°) had vector scores of 0 [$\cos(90°$ or $270°) = 0$]. By multiplying this value by each vector length, and summing over the trial duration, we quantified the female's relative attraction to the focal signal. A large positive net vector score indicated that the female moved quickly toward the focal speaker, a small positive score indicated the female moved slowly toward the focal speaker, a large negative score indicated the female moved quickly away from the focal speaker, and a small negative score indicated the female moved slowly away from the focal speaker (*Huber et al., 1984*).

### Open arena preference trials

We conducted the open arena trials in the same room and temperature conditions as spherical treadmill trials. We ran all trials in a chamber ($L \times W \times H = 111.7$ cm $\times 50$ cm $\times 25$ cm) with the walls lined with sound-attenuating foam. Speakers were located at opposite ends of the arena (lengthwise). We ensured the standard and focal signal would broadcast at the correct sound pressure level by pointing the sound level meter probe directly at the active speaker from the center of the arena, 55 cm from the active speaker, and setting the volume of the standard or focal signal accordingly. We demarcated a semicircle "choice zone" (radius = 28 cm) in front of each speaker. We then placed the female in the center of the arena (exactly 55 cm from each speaker) under an opaque plastic container ($L \times W \times H = 10$ cm $\times 8.5$ cm $\times 11$ cm). Prior to the start of data collection, the female experienced 60 s of silence, then 30 s of the focal sound and 30 s of the standard sound (order and speaker side randomized). We then broadcast both the focal and standard signals simultaneously, with the signal chirps interleaved. As soon as both

signals were being broadcast, we carefully and silently raised the opaque plastic container by pulling on its attached string. Data collection on female phonotaxis began as soon as the opaque container was removed.

Each trial ran for a total of 5 min. The following female responses were recorded: the amount of time spent stationary in the middle of the arena without moving after the opaque plastic container was removed, time spent in the arena outside of the choice zones (no-choice zone) after moving from the acclimatization location, time spent in focal choice zone, time spent in standard choice zone, and number of switches made between focal and standard choice zones. The base of the arena was vacuumed and wiped down after each trial.

Because of the varying amount of time spent in focal and standard choice zone, females were classified as 'preferring' a particular focal signal treatment if, over the course of the trial, she spent relatively more time in the zone with the focal signal than in the zone with the standard signal (i.e., time outside choice zones were not included to determine female choice) *sensu Bischoff, Gould & Rubenstein (1985)* and *Dugatkin & Godin (1992)*. To quantify female preference we calculated the proportion of time spent in the focal choice zone relative to the proportion of time spent in both choice zones combined. A number close to one indicates a strong phonotaxis response from the female with much more time spent in the focal zone than the standard zone; conversely, a number close to zero indicates much more time was spent in the standard zone than in the focal zone.

### Statistical analysis

All data were analyzed using JMP 10.0.0 statistical software (SAS Institute Inc., 100 SAS Campus Drive, Cary, North Carolina, USA). Female phonotaxis data (net vector scores from the spherical treadmill and proportion of time spent in focal zone from the open arena) were analyzed using repeated measures general linear mixed models (GLMM) that included sound pressure level, age, and the interaction between sound pressure level and age as fixed effects; body size (maximum head width) was included as a covariate; individual was treated as a random effect. To compare experimental methods, we converted all phonotaxis data into $z$ scores. $Z$ scores were then analyzed using a repeated measures GLMM. We included experimental method, sound pressure level, age, experimental method $\times$ sound pressure level, sound pressure level $\times$ age, experimental method $\times$ age, and experimental method $\times$ sound pressure level $\times$ age as fixed effects; body size was included as a covariate; we treated individual as a random effect. Because none of the interactions were significant in any of the models we re-ran all models without interactions per *Engqvist (2005)*. We only present the results of the partial models (excluding interactions) because partial F-tests revealed that the full models did not explain significantly more variation in female phonotaxis. We used one sample t-tests to determine which sound pressure levels invoked significantly different female phonotaxis levels relative to the standard. We used Tukey's post-hoc HSD tests to compare phonotaxis across different focal sound pressure levels.

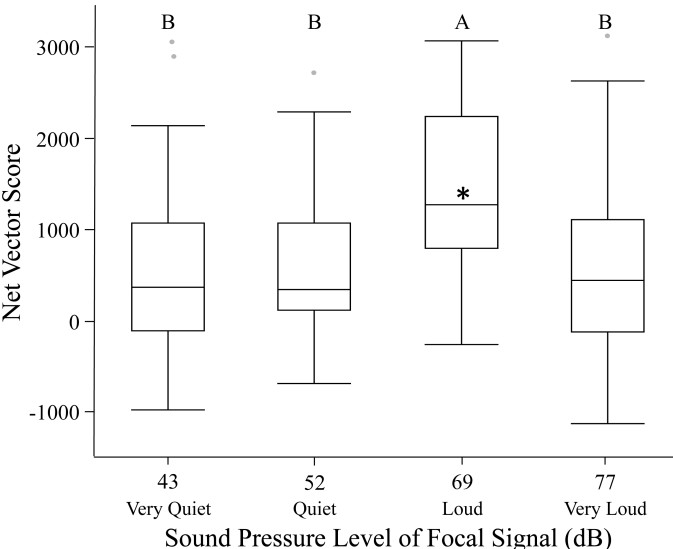

**Figure 1 Female phonotaxis under different sound pressure levels on the spherical treadmill.** Phonotaxis was quantified using net vector scores (net movement toward or away from the focal signal; direct movement towards the focal speaker resulted in positive vector scores while direct movement away from the focal speaker resulted in negative vector scores). Letters represent significant differences across sound pressure levels (Tukey's post-hoc HSD analysis). Asterisks represent significant differences between the focal and standard signal (above median = phonotaxis toward focal signal; below median = phonotaxis toward standard signal). Box plots: horizontal line within each box = median phonotaxis measure; top of each box = 3rd (75%) quartile phonotaxis measure; bottom of each box = 1st (25%) quartile phonotaxis measure; whiskers represent the outermost data point that falls within the distances 1st quartile − 1.5*interquartile range or 3rd quartile + 1.5*interquartile range; whiskers are determined by the upper and lower data point values (excluding outliers) when the data points do not reach the above computed ranges; light grey dots represent outliers, when present.

## RESULTS

Females in the spherical treadmill trials exhibited highly variable phonotaxis in response to the different focal sound pressure levels. Some females walked substantially farther than others, with distance covered ranging from 60 cm to 4,900 cm. Females had over a 60-fold difference in their velocity magnitudes, with the slowest moving at only 0.12 cm/s and the fastest moving at 8.22 cm/s. Females also moved in all directions during trials relative to the focal speaker. There was a 4-fold difference in net vector scores, indicating large variation in the strength of female phonotaxis on the spherical treadmill. Sound pressure level significantly influenced net vector scores, with females exhibiting highest phonotaxis towards the loud 69 dB $\cong \tilde{X} + 1$ SD SPL treatment relative to all other focal sound pressure level treatments (Fig. 1; Table 2). Age and size did not significantly influence female phonotaxis. Further, although individuals were included as a random effect in the GLMM, they did not explain any of the variation in phonotaxis. Post hoc one-way comparisons revealed that phonotaxis towards the loud 69 dB $\cong \tilde{X} + 1$ SD SPL was significantly higher than towards the standard 60.6 dB SPL ($t = 3.66$, $p = 0.0010$). Phonotaxis did not significantly differ relative to the standard sound pressure level in the other three focal treatments (all $p > 0.25$).

**Table 2 Repeated measures GLMM results for factors affecting female phonotaxis.** The spherical treadmill model used net vector scores to quantify phonotaxis and had an $R^2_{adj} = 0.0495$. The open model used proportion of time spent in focal choice zone relative to time spent in both focal and standard choice zones to quantify phonotaxis and had an $R^2_{adj} = 0.0356$. The method comparison model used $z$ scores to quantify phonotaxis and had an $R^2_{adj} = 0.0626$. Female phonotaxis was significantly influenced by sound pressure level but was not influenced by female age, female body size, or experimental method. Individual was identified as a random effect in all models, but did not explain any of the phonotaxis variation.

| Method | Source | DF | F | P |
|---|---|---|---|---|
| Spherical treadmill | Age | 3,84 | 0.8947 | 0.4475 |
| | Sound pressure level | 3,84 | 5.7035 | 0.0013 |
| | Body size | 1,28 | 0.1983 | 0.6595 |
| Open arena | Age | 3,79 | 0.7662 | 0.5164 |
| | Sound pressure level | 3,79 | 5.3900 | 0.0020 |
| | Body size | 1,29 | 0.3029 | 0.5864 |
| Both methods | Age | 3,192 | 0.7747 | 0.5094 |
| | Sound pressure level | 3,192 | 9.5844 | <0.0001 |
| | Technique | 1,193 | 0.0392 | 0.8433 |
| | Body size | 1,29 | 0.4584 | 0.5035 |

Females in the open arena trials also exhibited highly variable phonotaxis in response to the different focal sound pressure levels. In 17/120 of the open arena trials (14%) females failed to move out of the no-choice zone; these trials were excluded from analysis. Females usually spent time in both standard and focal choice zones in the remaining open arena trials. Sound pressure level significantly influenced female phonotaxis (relative time spent in focal choice zone). Females were most phonotactic towards the loud 69 dB $\cong \tilde{X} + 1$ SD SPL, relative to all other focal sound pressure level treatments (Fig. 2; Table 2). Age and size did not significantly influence proportion of time spent in focal choice zone. Further, individuals being included in the GLMM as a random effect did not explain any of the variation in phonotaxis. Post hoc one-way comparisons of the open arena data revealed that females spent significantly more time in the loud 69 dB $\cong \tilde{X} + 1$ SD SPL choice zone than the standard 60.6 dB SPL choice zone ($t = 2.47, p = 0.0197$). Conversely, when the sound pressure level was very loud 77 dB $\cong \tilde{X} + 2$ SD SPL, females avoided the focal choice zone, instead spending significantly more time in the standard 60.6 dB SPL choice zone ($t = -4.20, p = 0.0003$). Female time in the focal choice zone did not differ significantly relative to time in the standard sound pressure level choice zone in the quiet and very quiet treatments (all $p > 0.70$).

We used $z$ scores to compare phonotaxis across the two methods (spherical treadmill and open arena). Experimental method, age and size did not significantly influence female phonotaxis. Further, individuals being included in the GLMM as a random effect did not explain any of the variation in phonotaxis. Female phonotaxis was only influenced by sound pressure level, with females exhibiting highest phonotaxis towards the loud 69 dB $\cong \tilde{X} + 1$ SD SPL treatment relative to all other focal sound pressure level treatments (Fig. 3; Table 2).

**Peer**J ___________________________________________________________________

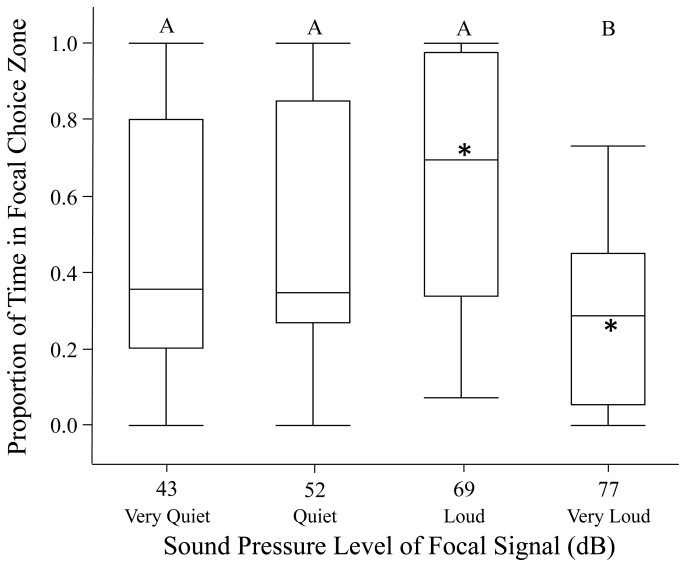

**Figure 2 Female phonotaxis under different sound pressure levels in the open arena.** Phonotaxis was quantified using proportion of time the female spent in the focal choice zone relative to the total amount of time spent in both the focal and standard choice zones. Box plot, asterisk, and letter descriptions provided in Fig. 1.

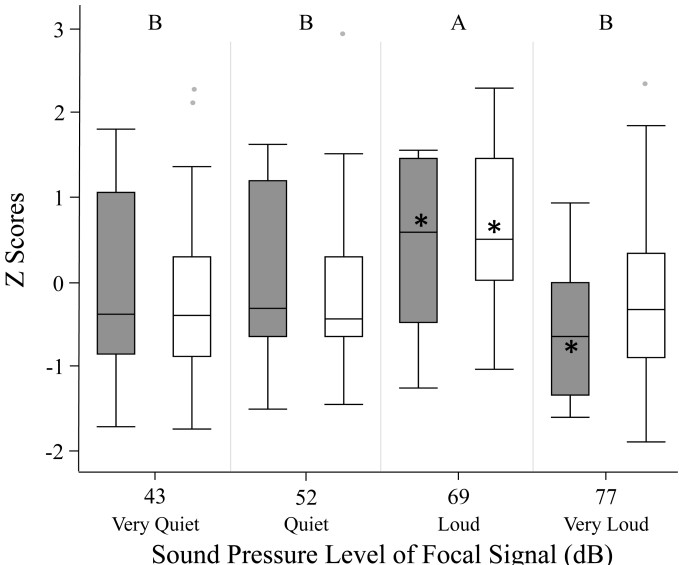

**Figure 3 Female phonotaxis under different sound pressure levels in both the spherical treadmill (white bars) and open arena (grey bars).** Phonotaxis was compared across the two experimental methods by converting all phonotaxis data into *z* scores. Box plot, asterisk, and letter descriptions provided in Fig. 1.

## DISCUSSION

The loud focal signal (69 dB $\cong \tilde{X} + 1$ SD SPL) invoked greater female phonotaxis than any other sound pressure level investigated (Figs. 1–3; Table 2). Overall, our findings suggest that males that signal at high sound pressure levels should experience a fitness advantage over males that signal at lower sound pressure levels, provided the sound pressure level is within the natural range of the species. This caveat is important, given females showed reduced phonotaxis towards the very loud focal signal (77 dB $\cong \tilde{X} + 2$ SD SPL), suggesting female preference for elevated sound pressure level is not open ended.

The very loud focal signal (77 dB $\cong \tilde{X} + 2$ SD SPL) may represent a novel supernormal stimulus as it falls beyond *G. assimilis*' natural range of 34–71 dB SPL (*Whattam & Bertram, 2011*). Many of the females exhibited negative phonotaxis when exposed to this supernormal stimulus (Figs. 1–3). *Weber, Thorson & Huber (1981)* showed a similar response in *Gryllus campestris* L, with females avoiding mate attraction signals above 70 dB SPL. This negative response to a supernormal stimuli has been observed previously, such as when female crickets exhibit negative phonotaxis to the number of syllables (pulses) in a chirp (*Hedrick & Weber, 1998*; *Ryan & Keddy-Hector, 1992*). Female negative phonotaxis to very loud signals may be an acoustic startle response (*Hoy, Nolen & Brodfuehrer, 1989*), or the females may have a perceived increase in predation risk (*Moiseff, Pollack & Hoy, 1978*; *Zuk & Kolluru, 1998*). It is also possible that absence of other sensory cues (*Crapon de Caprona & Ryan, 1990*) invoke this negative phonotaxis. In some species, such as *Xiphophorus nigrensis* and *X. pygmaeus* (Poeciliidae) swordtails, mate selection is dependent on the dual effect of both visual and olfactory cues.

Surprisingly, females did not exhibit significant phonotaxis towards the standard signal when the focal signal's sound pressure level was less than that of the standard signal (quiet 52 dB $\cong \tilde{X} - 1$ SD SPL or very quiet 43 dB $\cong \tilde{X} - 2$ SD SPL vs. 60.6 SPL; Figs. 1–3). These findings suggest females may not discriminate between sound levels unless one of the signals presents a higher than average sound pressure level. Regardless, given low sound levels may go undetected in nature due to elevated background noise levels, males should be selected to signal with high sound pressure levels whenever possible. It is unknown whether females can discern low sound pressure levels from the background noise level in the testing chamber ($42 \pm 2$ dB SPL); future tests should determine whether females exhibit phonotaxis towards very quiet (43 dB $\cong \tilde{X} - 2$ SD SPL) signals when the alternative is silence (*Brown & Handford, 1996*).

Our study compared open arena and spherical treadmill methods. Quantifying female preference for signal sound pressure level provides a powerful comparison of whether one experimental method quantifies female phonotaxis more effectively than the other because sound pressure level is one of the primary components that differ between the two methods. We found that the two methods produced virtually identical female phonotaxis results for almost all of the sound pressure levels examined. The only sound pressure level that evoked a slightly different phonotactic response between the two methods was the very loud treatment (77 dB $\cong \tilde{X} + 2$ SD SPL). Females in the open arena avoided the very

loud focal sound pressure level, spending significantly more time in the choice zone of the standard signal. While females in the spherical treadmill also exhibited somewhat reduced phonotaxis towards the very loud focal signal, phonotaxis did not significantly differ from the standard signal. Overall, our findings are consistent with the handful of other studies that compare these experimental methods. *Walikonis et al. (1991)*, *Stout, Atkins & Zacharias (1991)* and *Pires & Hoy (1992)* revealed female phonotaxis was the same in the open arena and on the spherical treadmill. Similar to our study, *Stout, Atkins & Zacharias (1991)* investigated female *Acheta domesticus* response to differing signal intensities. Conversely, *Walikonis et al. (1991)* investigated female *A. domesitcus* response to differing syllable periods while *Pires & Hoy (1992)* investigated female *G. firmus* response to natural calls recorded songs at different temperatures. Given sound pressure level, syllable period, and temperature are important predictors of mating preference, our joint findings suggest that open arena and spherical treadmill methods may be used interchangeably to quantify phonotaxis to acoustic signals.

Caution is warranted, however, when signals are broadcasted at unnaturally high sound pressure levels because of fundamental methodological differences between the open arena and the spherical treadmill. The open arena lets the female to move within the enclosed space, thereby enabling her to experience natural changes in sound pressure level. When individuals in the open arena run away from very loud signals, they experience a ∼6 dB drop in amplitude with every doubling of their distance from the very loud signal (assuming a spherical, radiating sound source without obstructions). However, when individuals on the spherical treadmill run away they do not experience a drop in amplitude regardless of the distance they run because they are tethered in place on the spherical treadmill. Future research should explore how important sound pressure level is relative to other signalling parameters.

## ACKNOWLEDGEMENTS

We thank three anonymous reviewers, along with Jeff Dawson, Jean-Guy J. Godin, Sarah J. Harrison, Genevieve L. Ferguson, and Michelle Loranger for helpful comments on an earlier version of this manuscript. We also thank Ryan Chlebak for designing and digitally rendering the spherical treadmill cowling, Michael Jutting for designing and rendering the electronics of the spherical treadmill, and Andrew Mikhail who completed preliminary tests of the device. Jeff Dawson was intimately involved with all aspects of designing and building the spherical treadmill and its accompanying software; this research could not have been conducted without him. We would like to thank Sarah J. Harrison, Genevieve L. Ferguson, Ian R. Thomson, and Kathryn Dufour for their help with cricket care. We would also like to thank Steven Gibson and the Stengl Lost Pines Biological Station at the University of Texas for hosting our laboratory during the cricket-collecting trip that resulted in the establishment of our *G. assimilis* laboratory population.

### Funding

Funding was provided to S.M.B. by a Natural Science and Engineering Research Council of Canada Discovery Grant, the Canadian Foundation for Innovation Grant, the Ontario Research Fund, and Carleton University Research Fund. The funders had no role in study design, data collection and analysis, decision to publish, or preparation of the manuscript.

### Grant Disclosures

The following grant information was disclosed by the authors:
Natural Science and Engineering Research Council of Canada Discovery Grant.
The Canadian Foundation for Innovation Grant.
The Ontario Research Fund.
Carleton University Research Fund.

### Competing Interests

Susan M. Bertram is an Academic Editor for PeerJ.

### Author Contributions

- Karen Pacheco conceived and designed the experiments, performed the experiments, analyzed the data, wrote the paper, prepared figures and/or tables, reviewed drafts of the paper.
- Susan M. Bertram conceived and designed the experiments, analyzed the data, contributed reagents/materials/analysis tools, wrote the paper, prepared figures and/or tables, reviewed drafts of the paper.

### Data deposition

The following information was supplied regarding the deposition of related data:
Figshare doi: http://dx.doi.org/10.6084/m9.figshare.1037378
http://figshare.com/articles/Male_Gryllus_assimilis_Standard_acoustic_attraction_signal/1037378
Title: Male (*Gryllus assimilis*) Standard acoustic attraction signal.

### Supplemental information

Supplemental information for this article can be found online at http://dx.doi.org/10.7717/peerj.437.

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
