# Peer review of "How male sound pressure level influences phonotaxis in virgin female Jamaican field crickets (Gryllus assimilis)"

_PeerJ, doi:10.7717/peerj.437_

## Round 0.1 · original submission · Minor Revisions

The three reviews all recommend accepting this MS, and two recommend acceptance with minor revisions. After reading the MS and the reviews, I agree with the majority.

All three reviews were provided by experts in the field, and all give excellent suggestions and provide important insight. I look forward to a comprehensive rebuttal and response from the authors.

I would also like to suggest that the authors deposit the electronic audio files used in these studies in a DOI-based database such as figshare or Dryad. The authors should then reference those digital objects in the revised MS. This would allow others to more precisely replicate these experiments.

Thank you for submitting your MS to PeerJ.

Reviewer 1 ·

Basic reporting

My main concern with this manuscript is that it does not distinguish between an active female preference for louder calls per se, versus a passive preference for closer males. Several times in the manuscript there is inadequate information on the standardization of dB by distance.

Line 26: do you suppose it is really correct that most insects rely on acoustic communication as stated? Many obviously do, but most?

Lines 89-91: sentence “All signal parameters….average male signal intensity” seems out of place; methods perhaps.

Lines 245, 255, 263: missing information “** % of the variation”

Figure 1: what is the horizontal line at about 750 net vector score?

Figure 2: what is the horizontal line slightly above zero? More importantly, this figure needs new Y-axis label and figure legend. It is not clear to have a negative value for time in “preferred zone” - if a zone gets less time, then it is not a preferred zone is it? The parenthetical “time in focal zone – time in standard zone” was the only way I could figure out what these data actually represent.

Figure 3: also has odd horizontal line…I don’t know what that represents.

Table 2: comparing methodologies section seems to have lost alignment (signal intensity wrapped text; age with no corresponding data). Also seems to have lost line of data (8 apparent variables, 7 rows of numbers). If the reason this is presented is to show that methodologies do not differ, then it is important that we can discern which row actually represents “Methodology”

Experimental design

Lines 130 – 137: standard broadcast was 60.6 dB @ 17 cm (line 130); the reference population average is given in line 133, but does not state at what distance that reference population recordings’ intensities were measured.

Lines 191 – 195: In the arena, the speakers were at opposite ends and the test cricket in the middle. This means that the crickets started about 55 cm from the speaker. It is not stated what sound intensities were used at 55 cm. Was it 60.6 db @ 55 cm standard, or was it 60.6 at 17 cm from the speaker?

Validity of the findings

Lines 266 – 269: the authors interpret their results to mean that females cannot differentiate quiet sounds (43, 52 dB) from background noise. This interpretation is not warranted given that all experiments were two speaker choice tests, i.e. there is never a test conducted to see if females would show phonotaxis to a 43 dB song if the alternative was silence.

Table 1: needs to standardize for distance, dB is meaningless unless it is dB at some distance. Ritchie 1992 Ephippiger is a katydid (Tettigoniidae) not a cricket (Gryllidae).

Reviewer 2 ·

Basic reporting

The language is often difficult to follow, for example in the abstract “the signal intensity of cricket male attraction signals”. ‘Of males’ or ‘of females’ would help to reduce such frequent clusters of adjectival nouns. The 3rd sentence is also difficult to understand: “- if females are simultaneously more attracted” Perhaps this means ‘if females see two males - - ? Vision is not mentioned, but isn’t it the only way a female could distinguish that he generates more sound pressure? Further in the abstract (barX+1SD= 69dB) and (barX+2SD=77dB) require the reader to calculate 61dB as the reference intensity, but it isn’t clear that both these methods (not methodologies surely) compare standard and experimental pressures.
Scientifically, the authors use intensity, loudness and pressure interchangeably but they are different. Intensity follows the square of pressure, but sound pressure is what is being measured and I believe should be used throughout. Loudness depends on the threshold frequency response of the listener and should definitely be avoided. The authors need to explain briefly focal sounds (L 119) females’ perceived predation risk and perceptual limits (Abstract) rather than just quoting previous work.

Experimental design

The design of the treadmill seems to prevent the female from orienting to the sound. It seems to me that the spring should be pivoted If she is to compare the output of the ears on her forelegs to approach a calling male. The design of the arena experiments is valid.

Validity of the findings

I find the conclusion “Females exhibited highest phonotaxis towards loud mate attraction signals” difficult to accept, particularly from the results of the treadmill experiments.

Additional comments

I see this as a draft of a shorter paper or note.

Reviewer 3 ·

Basic reporting

see General Comments below

Experimental design

see General Comments below

Validity of the findings

see General Comments below

Additional comments

The manuscript, “How male signaling intensity influences phonotaxis in virgin female Jamaican field crickets (Gryllus assimilis)” by Pacheco and Bertram is a very nice and straightforward experiment examining female preference for louder male acoustic signals in a fairly understudied species of field cricket. The paper is very well written, does a good job with the literature and is very clear in its methodology and conclusions. I don’t have any major concerns with it, and so restrict my review to pointing out specific things that could be tweaked to make an otherwise excellent paper even better.

lines 32-34: I’m not sure whether this is true “in general” – it depends on the kind of acoustic parameter being discussed. Certainly this is not the case for something like carrier frequency. The Ryan and Keddy-Hector paper is usually summarized as saying that females usually favour more energetic (louder, more persistent, etc.) signals.

lines 106-107: What time of day did the crickets’ light cycle start? This is important because later (line 154) you say that the phonotaxis trials were all done between 5 and 8pm, but the reader doesn’t know when during the crickets’ normal light cycle this is (I assume it fell during the night, but how far in would be nice to know and potentially an important factor).

line 107: How did you provide water? This question also applies to the later description of individual housing (line 115).

line 109: I suspect you mean haphazard rather than random. To be truly random you would have to have given all of the late instar females a unique id and then chosen 30 using a random number generator. It sounds like you did what most of us do – reach into the bin and pull out the first 30 animals that you see that have all their appendages intact. There’s nothing terribly wrong with this, but it shouldn’t be described as random.

line 134: Please explain why the -1 and -2 SD treatments are -10 and -19 dB below the mean (approx. 61 dB) when the measured population SD was 8 dB. Shouldn’t these treatments have been at 53 dB and 45 dB?

line 146: “in the dark under red light” sounds a bit like an oxymoron since if it was under red light then it certainly wasn’t dark. I suggest replacing this text with “under dim red light” and then giving the type and wattage of the red bulb used in parentheses.

line 147: Where were the females when they were acclimatizing? Were they left to roam freely in the arena or were they constrained to the middle somehow? If so, how might this have affected their hearing of the broadcasts, or does it matter? On the treadmill, were females free to walk while acclimatizing, or was the ball fixed? [Oh, I see, this information comes later.]

line 172: There seems to be some inconsistency with the temperatures at which the various parts of this study were conducted. The crickets were raised at 28C, the male recordings were taken at 26C and the phonotaxis trials were done at 22-23C. There is plenty of literature to show that structural components of male song change with temperature, and although I don’t know of any studies showing that female preferences change with temperature – I would bet they do. I don’t really see the above variation as a serious flaw, but it is puzzling why temperature wasn’t controlled across the various components (those portable, oil-filled heaters can do wonders in a pinch). Do you have an explanation?

line 227: I think there is an extra symbol in this formula (upside down question mark).

line 230: The analysis as it stands can detect a main effect difference between the two phonotaxis methods, but not whether females change the way they respond (i.e. preference shape), in other words whether there is an interaction between method and response. I suspect that you don’t have enough degrees of freedom to do this, but the best way to see if there is a difference between the methods is to build a model with all interactions between test type and the treatment effects and see if that interaction is significant. You could also conduct a partial F-test between this fully-specified model (with interaction) and the reduced model (without) – I think this is called a sequential model building approach (e.g. see Chenoweth and Blows 2005. Contrasting mutual sexual selection on homologous signal traits in Drosophila serrata. Am Nat 165: 281-289). [I see that you actually did test for an interaction later in the results (lines 261-263) – perhaps this could be mentioned here?]

lines 242-243: Were the responses significantly reduced relative to the standard (i.e. were the scores significantly below 0) or just significantly below the +1 SD treatment? I don’t think the analysis as presented so far is equipped to test the former as that would require a simple one-sample test for each treatment. I would encourage the authors to do such tests as they would answer the question on most readers’ minds as to whether each dB treatment level resulted in attraction or repulsion (not simply whether treatments differed as is presented here).

line 245: I think the authors have forgotten to replace a placeholder “**” with an actual value (also at lines 255 and 263).

line 247-249: Why exclude them, couldn’t they just be recorded as zero? This would balance your design and bring it more in line with the treadmill test where undoubtedly some of the slow-moving females would never have come close to any sound source or even moved far enough to detect a change in source intensity.

lines 251-253: This result implies a test relative to the zero line of no preference – was that done?

lines 261-263: This partially takes care of my earlier comment, but were the interaction terms removed after they were found to be not significant? This should be done otherwise the tests of main effects are not valid (see Engqvist 2005. The mistreatment of covariate interaction terms in linear model analyses of behavioural and evolutionary ecology studies. Anim Behav 70: 967-971).

line 267: As I’ve mentioned above, these statements would require tests for differences between each treatment and the zero mark.

lines 269-273: There is a reasonably large literature out there looking at masking of sound by background noise and what one might predict would be the results. I would suggest integrating some of that literature into your discussion. Try looking for papers by Paul Handford or XXX.

line 271: “may go undetected” at further distances from the source.

line 309: Not louder absolutely, but louder within the natural range of male signal intensities. I suggest that you qualify the statement.

line 314: Are there any other examples of female preferences that are strongly linear within the natural range of signals drop off when stimuli are supernormal? I suggest trying to find some other examples as this will help the reader put these (potentially novel) results into context. I can’t think of any off the top of my head. The supernormal stimuli that come to mind (Malte Andersson’s extension of widowbird tails, or Nancy Burley’s adding novel ornaments to zebra finch plumage) all seem to suggest that female preferences for ornaments should be open-ended.

line 333: What do the upper and lower bounds of the box indicate (I assume the upper and lower quartiles)? Do you mean “median” here when you say “mean”?

Figure 1: I would change the y-axis title to simply read, “Vector Score” or, “Net Vector Score” since the values for each female do not represent averages, neither are averages displayed on the graph.

Figure 2: The current y-axis title, “Average Time Spent in Preferred Choice Zone (sec)” is not an accurate description of this variable and it isn’t consistent with the figure legend (e.g. “relative” is used there instead of “average”). A more accurate title would be, “Net time spent in focal choice zone (focal – standard, s)”.

Figures 1-3: I would consider replacing the x-axis labels -2, -1, etc. with the actual dB measurements.

Table 1: Although Lickman et al. 1998, Prosser et al. 1997 and Wagner et al. 1995 are nominally about Gryllus integer, they were actually working with G. texensis. Since 2000 when the species was formally described, populations of G. integer in Texas are now recognized as being G. texensis. As you can see from the data, they are much closer to the G. texensis in Gray and Cade 2000 than to the true G. integer (which is found in California). I’m no taxonomist, but I know there’s some specific way of indicating that the name has changed for those papers.

Table 2: Can you explain the degrees of freedom as listed? Why are there so many and why are they so varied? Also, where is female identify as a random factor? Finally, in the bottom section on “Comparing Methodologies” the alignment seems to be a bit messed up as well as there being no rules.

---

## Round 0.2 · accepted · Accept

Thank you for completing the requested revisions in a comprehensive manner, and for your detailed rebuttal letter. Thanks, too, to the three anonymous reviewers who provided useful suggestions and constructive criticism.

Please consider publishing the reviews and your response alongside your article, as it adds value to the study both in terms of important background information and for use in educational and mentoring contexts.

Thanks again for submitting your research work to PeerJ.